# In Vitro Study of the Multimodal Effect of Na^+^/K^+^ ATPase Blocker Ouabain on the Tumor Microenvironment and Malignant Cells

**DOI:** 10.3390/biomedicines11082205

**Published:** 2023-08-05

**Authors:** Octavia-Oana Harich, Oana-Isabella Gavriliuc, Valentin-Laurentiu Ordodi, Alexandru Tirziu, Virgil Paunescu, Carmen Panaitescu, Maria-Florina Bojin

**Affiliations:** 1Department of Functional Sciences, Immuno-Physiology and Biotechnologies Center, “Victor Babes” University of Medicine and Pharmacy, No. 2 Eftimie Murgu Square, 300041 Timisoara, Romania; harich.octavia@umft.ro (O.-O.H.); valentin.ordodi@upt.ro (V.-L.O.); alexandru.tirziu@umft.ro (A.T.); vpaunescu@umft.ro (V.P.); cbunu@umft.ro (C.P.); florinabojin@umft.ro (M.-F.B.); 2Faculty of Industrial Chemistry and Environmental Engineering, “Politehnica” University Timisoara, No 2 Victoriei Square, 300006 Timisoara, Romania; 3Center for Gene and Cellular Therapies in the Treatment of Cancer Timisoara-OncoGen, Clinical Emergency County Hospital “Pius Brinzeu” Timisoara, No. 156 Liviu Rebreanu, 300723 Timisoara, Romania

**Keywords:** ouabain, Na^+^/K^+^ ATPase, SK-BR-3 tumor cells, TAFs, MSC, molecular target

## Abstract

Na^+^/K^+^ ATPase is a protein involved in the active transport of ions across the cellular membrane. Ouabain is a cardiotonic glycoside that, by inhibiting the Na^+^/K^+^ pump, interferes with cell processes mediated directly by the pump, but also indirectly influences other cellular processes such as cell cycle and proliferation, growth, cell differentiation, angiogenesis, migration, adhesion, and invasion. We used the SK-BR-3 breast cancer cell line, mesenchymal stem cells (MSCs), and tumor-associated fibroblasts (TAFs) in vitro to determine the effects of ouabain exposure on these cellular types. The results showed a multi-level effect of ouabain mainly on tumor cells, in a dose-dependent manner, while the TAFs and their normal counterparts were not significantly influenced. Following exposure to ouabain, the SK-BR-3 cells changed their morphologic appearance, decreased the expression of immunophenotypic markers (CD29, Her2, VEGF), the proliferation rate was significantly decreased (Ki67 index), the cells were blocked in the G_0_ phase of the cell cycle and suffered necrosis. These data were correlated with the variable expression of α and β Na^+^/K^+^ pump subunits in tumor cells, resulting in decreased ability to adhere to the VCAM-1 substrate in functional flow chamber studies. Being indicative of the pro-apoptotic and inhibitory effect of ouabain on tumor invasion and metastasis, the results support the addition of ouabain to the oncological therapeutic arsenal, trailing the “repurposing drugs” approach.

## 1. Introduction

The sodium–potassium pump is a transmembrane ATPase that is involved in the active transport of Na^+^ and K^+^ ions across the cell membrane and thus contributes to the maintenance of the relative electronegativity of the intracellular space, the resting transmembrane electrochemical gradient as well as the osmolarity of the intracellular and interstitial environment, with crucial roles for important physiological processes such as the production of neuronal action potential, the filtering of waste products in the nephrons, or sperm motility [1]. In addition, Na^+^/K^+^ ATPase interacts with a number of membrane and intracytoplasmic proteins, intervening in the processes of cell growth, differentiation, or apoptosis [1,2].

On account of its involvement in the regulation of several cellular functions, inhibition of the Na^+^/K^+^ pump can potentially have many useful biological consequences and has thus been successfully exploited in the development of cardiovascular medication [3,4]. Ouabain belongs to the widely distributed family of plant-derived cardiotonic glycosides, which, by inhibiting the Na^+^/K^+^ pump, causes an increase in intracellular Na^+^ ions, which further increases the concentration of Ca^2+^ ions and consequently exerts positive inotropic effects [5,6]. Secondary to an intracytoplasmic Ca^2+^ increase, cascade activation of calmodulin, calmodulin kinase, and Cdc25 phosphatase occurs, which results in the subsequent inhibition of cyclin-dependent kinase 1 and cyclin B. Blocking cyclin B results in cell cycle arrest. In addition, a reduction in cyclin D levels was observed, resulting in a blockade of the cell cycle in the G_0_/G_1_ phase [6]. 

In addition to ion transporter activity, the Na^+^/K^+^ pump is involved in ion-independent cell signaling processes [7,8,9,10]. One of the most important interactions of the pump is between the alpha-1 chain and the Src protein. This interaction keeps Src in an inactive state and implicitly inhibits any Src-dependent process [11]. Once ATPase is blocked, Src dissociates and becomes active. Following this activation, Src transactivates the EGFR, increases the production of reactive oxygen species (ROS), and activates transcription factors such as NF-κB [12]. Additionally, Na^+^/K^+^ pump blockage can lead to the activation of the Ras and JAK-STAT signaling pathways, which are involved in the processes of growth, cell differentiation, angiogenesis, migration, adhesion, and invasion [10,13].

Due to the antiproliferative and cytotoxic potential, the repurposing of cardiac glycosides for cancer treatment has recently gathered significant interest [14]. Interestingly, cardiac glycosides have also been reported to be inducers of immunogenic cell death, which promote the immunological clearance of cancer cells and increase the efficacy of antitumoral interventions by modulating the activity of T cells [15].

In view of expanding the oncological therapeutic arsenal, this study aims to provide further insights into the effect that blocking the Na^+^/K^+^ pump with ouabain would have in the tumor microenvironment by investigating changes in the proliferation, adhesion, and expression of phenotypic and molecular markers of tumor cells and tumor-associated cells compared to normal cells. 

## 2. Materials and Methods

### 2.1. Cell Cultures, Reagents, and Solutions

The SK-BR-3 breast cancer cell line was purchased from American Type Culture Collection (ATCC, HTB-30TM) and further expanded in McCoy’s 5A medium (Gibco BRL, Invitrogen, Carlsbad, CA, USA; Cat. no. 16600-082), supplemented with 10% fetal calf serum (FCS, PromoCell, Heidelberg, Germany; Cat. no. C-37350) and 100 IU/mL penicillin/100 µg/mL streptomycin solution (Pen/Strep, 10,000 IU/mL; PromoCell; Cat. no. C-42010). 

Normal human bone marrow derived mesenchymal stem cells (hMSCs; Cat. no. PT-2501) were purchased from Lonza (Basel, Switzerland) and further cultured and expanded in alpha-minimum essential medium (MEM, Gibco BRL; Cat. no. 22561-021) containing 10% FCS (PromoCell) and 100 IU/mL penicillin/100 µg/mL streptomycin solution (PromoCell).

Human tumor-associated fibroblasts (TAFs) were accessed from the Oncogen Biobank, being previously isolated using the method described by Paunescu et al. [16]. Briefly, breast cancer tissue-dissociated cells were washed several times with phosphate-buffered saline (PBS, Gibco; Cat. no. 14190-094) solution, passed through 0.70/0.40 μm strainer filters, plated as a single-cell suspension in adherent plastic culture plates, and further expanded in Dulbecco’s modified Eagle medium (DMEM; Sigma-Aldrich Company, Saint Louis, MO, USA; Cat. no. D0822) supplemented with 10% FCS (PromoCell) and 200 IU/mL pen/200 µg/mL strep solution (PromoCell). 

All cellular types were grown at 37 °C in a humid atmosphere containing 5% CO_2_. The medium was replaced every 2–3 days until reaching 80–90% confluence, when the cells were detached from the culture flasks using 0.25% Trypsin-EDTA solution (Sigma-Aldrich Company; Cat. no. T4049), followed by centrifugation (10 min, 300× *g*) and re-plated in appropriate culture flasks at a density of 10,000 cells/cm^2^. 

Ouabain octahydrate was purchased from the Sigma-Aldrich Company as >95% (HPLC) powder (Cat. no. 11018-89-6) and a stock solution of 12 × 10^−5^ M was prepared using PBS (Gibco). The concentrations of 10^−5^ M, 10^−6^ M, 10^−8^ M, and 10^−9^ M were obtained in culture media specific for each cellular type, and were calculated based on the minimum and maximum concentrations reached in biodistribution studies, so that the volume of ouabain solution (in PBS) would not influence the composition of nutrients in the culture media. 

### 2.2. Immunocytochemistry

Immunocytochemistry was performed for the MSCs, TAFs, and SK-BR-3 cell lines. Cells prepared for these analyses were grown in 4-well chamber slides for 24 h (control) or for an additional 24 h with different ouabain concentrations in the culture media; cells were washed, fixed with 4% paraformaldehyde, permeabilized with 0.1% Triton X-100 (Sigma-Aldrich; Cat. no. 9036-19-5) in PBS buffer, and incubated for 15 min at room temperature in the dark. Proteins of interest were further investigated by labeling them with the following antibodies: monoclonal mouse anti-swine vimentin (clone V9; Cat. no. A0485), monoclonal anti-human endoglin, CD105 (clone SN6h; Cat. no. M3527), mono-clonal mouse anti-human vascular endothelial growth factor, VEGF (clone VG1; Cat. no. M7273), monoclonal mouse anti-human Ki67 (clone MIB-1; Cat. no. M7240), polyclonal rabbit anti-human CD117 (Cat. no. A4502), polyclonal rabbit anti-human c-erbB-2 oncoprotein (Cat. no. A0485). These antibodies were provided by DakoCytomation (Glostrup, Denmark) and tested for human specificity and cross-reactivity. For the identification of Na^+^/K^+^ pump subunits, we used monoclonal mouse sodium potassium ATPase Alpha 1 (clone 464.6; Cat. no. NB300-146) and Beta 1 (clone 464.8; Cat. no. NB300-147) antibodies (Novus Biological). The staining protocol continued with secondary biotinylated antibody binding, substrate addition, and hematoxylin (Dako; Cat. no. CS70030-2) counterstaining of the nuclei (LSAB2 System-HRP, Cat. no. K0675, and Envision Kit, Cat. no. K5007, Dako) following the manufacturer’s procedures. Microscopy visualization was performed on a Nikon Eclipse E800 microscope (Nikon, Tokyo, Japan) and ImageJ (version 1.53t) was used for further analysis [17].

### 2.3. Flow Cytometry 

MSCs and TAFs were detached from the culture flasks using 0.25% Trypsin-EDTA (Sigma; Cat. no. T4049), washed twice with PBS, resuspended in 100 µL PBS at a con-centration of 10^5^ cells/mL, and incubated in the dark at room temperature for 30 min with a mouse anti-human fluorochrome-conjugated antibody at a dilution specified in the manufacturer’s protocol. Cells were then washed twice with 1 mL Cell Wash Solution (BD Biosciences, San Jose, CA, USA) each and resuspended in 500 μL of the same solution for further analysis on a four-color capable FACSCalibur (BD Biosciences) flow cytometer. Antibody panels used for these cellular types included PE-conjugated mouse anti-human CD29 (clone MAR4; Cat. no. 557332), CD73 (clone AD2; Cat. no. 550257), and FITC-conjugated CD90 (clone 5E10; Cat. no. 561969). Acquisition and data analyses were performed using CellQuest Pro software (BD, version 5.1). SK-BR-3 cells were subjected to the same procedures using FITC mouse anti-human Her-2/neu (clone NEU 24.7; Cat. no. 340553), CD29-PE, and PE mouse anti-human CD309 (VEGFR-2; clone 89106; Cat. no. 560494. All antibodies were purchased from BD Pharmingen™, BD Biosciences. Flow cytometric analyses were performed for control cells and for cells treated with different ouabain concentrations (10^−5^ M, 10^−6^ M, 10^−8^ M, 10^−9^ M) for 24 h.

### 2.4. Annexin V/PI Assay

Annexin V-FITC (Miltenyi Biotec, Gladbach, Germany; Cat. no. 130-092-052) was used in the cell death flow cytometric studies (apoptosis) combined with Propidium Iodide Staining Solution (BD Biosciences; Cat. no. 556463) following the manufacturer’s protocol. In brief, 10^6^ cells were washed with 1 × Annexin V Binding Buffer (BD Pharmigen™) and centrifuged at 300× *g* for 10 min, resuspended in the same solution, and incubated with 10 μL of Annexin V-FITC for 15 min in the dark. Washing the cells with 1 mL specific binding buffer and centrifugation were the next steps in the procedure, and the cell pellet was resuspended in 500 μL binding buffer while 1 μg/mL of PI solution was added immediately prior to analysis by flow cytometry. All cellular types were analyzed as the control and 24-h after adding the ouabain solution at concentrations of 10^−5^ M, 10^−6^ M, 10^−8^ M, and 10^−9^ M.

### 2.5. Cell Cycle Test

We used the CycleTESTTM PLUS DNA Reagent Kit (BD; Cat. no. 340242), which provides a set of reagents for nuclear isolation and labeling from cell suspensions. In this study, cell cycle testing was required to identify the phases in which cells were affected by 24-h exposure to different ouabain concentrations (10^−5^ M, 10^−6^ M, 10^−8^ M, and 10^−9^ M). This method involves the dissolution of lipids from cell membranes with the help of non-ionic detergents, the elimination of the cytoskeleton and nuclear proteins using trypsin, and chromatin is stabilized with spermine. Propidium iodide (PI) is bound stoichiometrically to the isolated nuclei, the samples being acquired by a flow cytometer. Nuclei labeled with PI emit fluorescence at wavelengths between 580 nm and 650 nm. The FL2 detector of the flow cytometer (FACSCalibur, BD Biosciences) was used to detect the fluorescence emission at wavelengths between 564 nm and 606 nm by the labeled cells. Solution A—trypsin/spermine tetrachloride, the addition of 250 μL, and incubation for 10 min at room temperature; Solution B—trypsin inhibitor/ribonuclease A to inhibit the trypsin activity and RNA, followed by the addition of 200 μL and incubation for 10 min at room temperature; Solution C—propidium iodide, binds to DNA; the addition of 200 μL was followed by flow cytometric analysis.

### 2.6. Flow Chamber Assay

A 6-channel μ-Slide VI ibiTreat flow chamber (Ibidi Integrated BioDiagnostics, Mu-nich, Germany; Cat. no. 80626) was coated with VCAM-1 (vascular cell adhesion pro-tein-1; R&D Systems; Cat. no. 809-VR) at a concentration of 2 μg/mL (in sterile deionized water), 30 μL/channel, 15 min before the experiment. Untreated and 24-h ouabain-treated SK-BR-3, TAFs, and MSCs were suspended in HBSS medium (Hank’s Balanced Salt Solution, Gibco; Cat. no. 14025092), 10^5^ cells/100 μL medium/channel in order to test their capacity to adhere to the VCAM-1 substrate under progressively increasing shear stresses of 0.35, 2, 5, 8, and 15 dynes/cm^2^ generated using an ISMATEC pump—IPC High Precision Multichannel Dispenser (IDEX Corporation, Glattburgg, Switzerland). Pictures of the centered microscopic field were taken every 30 s for every value of shear stress, and the total cell count was compared with the control. Variations of at least 15% in cell count were considered significant when compared with the control cells for the same values of the shear stress. 

### 2.7. RNA Extraction and RT-PCR

The total RNA was extracted with TRIzol reagent (Invitrogen™, ThermoFisher Scientific; Cat. no. 15596026) following the supplier’s instructions. RNA concentration was determined with an ND-1000 spectrophotometer (Wilmington, DE, USA). We used 0.5 μg of total RNA for each reverse transcription reaction performed with the AccuScript High Fidelity 1st Strand cDNA Synthesis Kit (Stratagene, Agilent Technologies, USA; Cat. no. 200820). The cDNA samples were analyzed by quantitative real-time PCR using the LightCycler 480 SYBR Green I Master (Roche, Florence, SC, USA; Cat. no. 04707516001), and the primers are listed in Table 1.

HPRT1 was chosen as a suitable reference gene. We performed a relative basic quantitation based on the ΔΔCt method with the LightCycler480 Software (Roche, version 1.2.9.11). Quantitative analysis of gene expression was performed for the α1, α2, and β1 subunits. For the other subunits, only semi-quantitative analysis was performed by RT-PCR, followed by visualization in 1.5% agarose gel.

### 2.8. Electron Microscopy

Scanning electron microscopy (SEM) was performed for the identification of ouabain-induced morphological changes in the MSCs, TAFs, and SK-BR-3 cell line. Cells were cultured at 10,000 cells/cm^2^ in 24-well format cell culture inserts (BD Labware Europe, Le Pont De Claix, France; Cat. no. 353095), and ouabain was added in concentrations of 10^−5^ M, 10^−6^ M, 10^−8^ M, and 10^−9^ M. After 24 h, the cells were pre-fixed for 1 h with 2.5% buffered glutaraldehyde (in PBS), rinsed three times in PBS, and the 0.4 μm pore-sized membranes were detached from the culture inserts. Better image quality was obtained after the cells were sputter-coated with platinum–palladium; SEM analysis was performed with an FEI Quanta 3D FEG electron microscope (FEI Company, Eindhoven, The Netherlands).

### 2.9. Statistical Analysis

Statistical analysis was performed using Excel Microsoft Office 2003 software (Microsoft Corporation, Redmond, WA, USA). The Student’s *t*-test and ANOVA were conducted for continuous variables. The values were expressed as the mean (M) ± standard deviation (sd). Differences were considered significant for *p* < 0.05.

## 3. Results

### 3.1. Morphological Changes of Ouabain-Treated Cells

The analysis of the scanning electron microscopy (SEM) images revealed that the SK-BR-3 cancer cells that demonstrated resistance to the cytotoxic effect of ouabain in concentrations of 10^−^^5^ M or 10^−^^6^ M became more adherent to the culture substate, acquiring characteristics of “ghost cells” (Figure 1A). MSCs also increased their area of contact with the culture substrate, enlarging their volume, whereas the nucleus became indented (Figure 1B). TAFs, on the other hand, underwent only minor morphological changes when subjected to 10^−^^6^ M ouabain for 24 h (Figure 1C).

### 3.2. Ouabain-Induced Immunophenotypic Changes in Tumor, Tumor-Associated Cells, and MSCs

Ouabain-induced changes in the expression of several surface markers were assessed by flow cytometry for all ouabain concentrations 24 h after addition to the culture media. The SK-BR-3 breast cancer cells showed a decrease in the expression of VEGF-R2 (Figure 2), while Her2 and CD29 presented insignificant variation compared to the control, untreated cells. The expression of characteristic stromal surface markers CD90, CD29, and CD73 was more severely affected by ouabain treatment in the MSCs and TAFs, resulting in a more than 70% decrease (Figure 2).

The effect of ouabain treatment on typical cellular types that can be encountered in the tumor niche, mainly a reduction in the expression of characteristic and cell proliferation markers, was investigated using immunocytochemistry (Figure 3).

Expression of vimentin, Ki67, c-kit, and endoglin (CD105) was determined using immunocytochemistry 24 h after the exposure of the MSCs and TAFs to a concentration of 10^−6^ M ouabain in the culture medium. A decrease in the expression of the nuclear proliferation marker Ki67, which was expressed in 25% of the cells subjected to the action of the Na^+^/K^+^ pump blocker, was indicative of a reduced proliferation rate for MSCs and TAFs. The other markers did not present significant variations in expression levels in either the MSCs or TAFs (Figure 3A). 

The average value for Ki67 expression in the population of untreated SK-BR-3 cells was 61.68%, whereas in the ouabain-treated cells, Ki67 was expressed only in 19.31%, thus suggesting that a concentration of 10^−6^ M of the Na^+^/K^+^ pump blocker has an antiproliferative effect on SK-BR-3 tumor cells. Furthermore, the cell line characteristic marker, epithelial growth factor receptor Her2, was decreased in the ouabain-treated cells as well as in the intracytoplasmic secretion of vascular endothelial growth factor (VEGF) (Figure 3B). The endoglin (CD105) marker was not expressed on the SK-BR-3 cells (Appendix A).

### 3.3. Effect of Ouabain on Cellular Viability and Cell Cycle

Following flow cytometric analysis with Annexin V and PI, an increase in the percentage of apoptotic SK-BR-3 cells (positive for Annexin V, negative for propidium iodide) was observed, ranging from 6.15% (control group) to 75.67% (cells exposed to 10^−5^ M ouabain, for 24 h). The increase in the number of apoptotic cells occurred in a dose-dependent manner, suggesting that ouabain exerts a pro-apoptotic effect on SK-BR-3 tumor cells (Figure 4).

Ouabain slightly reduced the viability of MSCs in a concentration-dependent manner, with the most significant impact seen at a concentration of 10^−6^ M, where 7.74% of cells showed PI+ staining. On the other hand, the impact of ouabain treatment on the viability of TAFs was less significant, showing only 2.66% PI+ cells when exposed to 10^−6^ ouabain. 

The MTT-based cytotoxicity assay showed similar results regarding the cell viability after different concentrations of ouabain treatment (Appendix A). 

Following exposure to different ouabain concentrations, a corresponding dose-dependent increase in the number of apoptotic SK-BR-3 cells (in the sub-G_0_ phase) and a decrease in the number of cells in the synthesis phase (S) were observed. The percentage of cells in the G_0_/G_1_ and G_2_/M phases remained relatively constant. A noteworthy observation was that a sub-G_0_ phase was observed in approximately 10–15% of the SK-BR-3 cells when exposed to high concentrations (10^−6^ M, 10^−5^ M) of ouabain (Figure 5A). 

While all concentrations of ouabain used in the experiment led to the arrest of MSCs in the G_0_/G_1_ phase of the cell cycle, the Na^+^/K^+^ pump blocker exerted a minimal influence on TAFs (Figure 5B,C). Cell cycle data determined by flow cytometry were correlated with the viability/apoptosis Annexin/PI test performed under the same experimental conditions (Figure 4).

### 3.4. Functional Studies—Flow Chamber Assay

The SK-BR-3 cells, exposed to various concentrations of ouabain (10^−^^5^ M, 10^−^^6^ M, 10^−^^8^ M, and 10^−^^9^ M), were placed in a VCAM-1 coated flow chamber and subsequently subjected to increasing shear stresses from 0.35 dyne/cm^2^ up to 15 dyne/cm^2^. Starting with 0.35 dyne/cm^2^, a significant decrease (>15%) in cell adhesion was observed in the cells treated with 10^−^^8^ M ouabain, while for cells treated with 10^−^^5^ M and 10^−^^6^ M ouabain, a comparative effect was obtained following exposure to shear stresses of 2.5, 8, and 15 dynes/cm^2^ (Figure 6A).

The ability to retain adherence to the VCAM-1 substrate under increasing shear stresses was similar for both the MSCs and TAFs, which decreased significantly for all of the shear stresses applied, but only for the cells treated with high concentrations of ouabain (10^−^^5^ M and 10^−^^6^ M) (Figure 6B,C).

These observations suggest the existence of an ouabain-mediated process by which tumor cells reduce the number of protein interactions with the VCAM-1 molecules lining the flow chamber. Reduced protein interactions with VCAM-1 may be attributed to the altered expression of adhesion molecules or a cytoskeletal change with an effect on cell geometry.

### 3.5. Effects of Ouabain on the Expression of the Na^+^/K^+^ Pump Subunits

Quantitative analysis of the relative gene expression of the α1, α2, and β1 subunits of the Na^+^/K^+^ pump was performed using the ΔΔCt method and HPRT1 reference gene. The results revealed major changes induced by exposure to ouabain in the gene expression of the Na^+^/K^+^ pump subunits. Thus, the expression of the α1 subunit showed little change for low concentrations of ouabain, but a significant increase for concentrations of 10^−^^6^ and 10^−^^5^ M. The MSCs showed the most prominent increase in the expression of the α1 subunit of Na^+^/K^+^ pump for treatment with 10^−^^6^ ouabain, but at very high concentrations of ouabain (10^−^^5^ M), its expression decreased compared to the values determined in the SK-BR-3 cells (Figure 7A).

The α2 subunit of the Na^+^/K^+^ pump was highly expressed in tumor cells when compared to both MSCs and TAFs, with an overexpression for the 10^−^^5^ M ouabain concentration (Figure 7B). 

The β1 subunit showed an increased gene expression following exposure to ouabain concentrations between 10^−^^9^ and 10^−^^6^ M for both MSCs and TAFs, whereas for the highest ouabain concentration (10^−^^5^ M), its expression decreased, showing comparable values for all cell types analyzed (Figure 7C). Knowing that the β1 subunit of the Na^+^/K^+^ pump, responsible for anchoring the Na^+^/K^+^ ATPase complex to the membrane, is downregulated, it could lead to a decrease in the number of Na^+^/K^+^ pumps found on the cell membrane. 

Agarose gel visualization of the other subunits and semi-quantitative analysis (using ImageJ, Analyze Gels) showed that MSCs had the highest stability of gene expression for the α3, β2, and β3 subunits, while the TAFs expressed these subunits in low quantity, also being downregulated when ouabain was added to the culture medium in progressively increasing concentrations. The SK-BR-3 cells expressed significantly lower β2 and β3 subunits when treated with high concentrations of ouabain (10^−^^6^ and 10^−^^5^ M) compared with the untreated cells (Figure 7D and Appendix A). 

At the protein level, the expression of the α1 and β1 subunits showed the same expression pattern, highlighted in gene determinations (Figure 8A,B).

## 4. Discussion

The oncological therapeutic arsenal is constantly expanding but still remains incomplete. This present gap is explained by the incomplete knowledge of all the mechanisms underlying tumorigenesis, tumor invasion, and metastasis. Cardiotonic glycosides such as ouabain are characterized by the ability to block the ion pump activity of Na^+^/K^+^ ATPase, whose varied roles in cell growth, proliferation, differentiation, and adhesion have recently been described, consequently inducing various changes in the cellular phenotype, from the expression of structural proteins such as adhesion molecules or cytoskeletal proteins, to that of functional proteins involved in cell proliferation and metabolism [18,19].

Reduction in cell viability following ouabain treatment, as assessed by the Annexin V/PI assay and by immunocytochemistry using the Ki67 index, was observed in the SK-BR-3 cells in a concentration-dependent manner. The suppression of ATPase function by ouabain resulted in a decrease in the internalization of nutrients necessary to maintain proper cellular metabolism. The blockage of the Na^+^/K^+^ pump generates a pathological loop whereby the reduction in the internalized glucose levels will lead to a depletion of ATP generated at the mitochondrial level and implicitly, a reduction in the Na^+^/K^+^ pump activity through ATP deficiency. The results obtained are consistent with the experiments performed by Yang et al. on U-87MG glioma cells [20], Xiao et al. on OS-RC2 renal carcinoma cells [21], Ninsontia et al. on H460 lung cancer cells [22], de Souza et al. on Caco-2 colorectal cancer [23], Chang et al. on DU-145 prostate cancer cells [10], Khajah et al. on MCF-7, YS1.2, pII, and YS2.5 breast cancer cells [11], and on osteosarcoma and squamous cell carcinoma [24,25]. The results of the Annexin V/PI investigation suggest that the majority of SK-BR-3 cells treated with cardiotonic glycosides do not undergo a process of necrosis, but die by apoptosis. These results support the data obtained by Da Silva et al. [26] on murine Tregs lymphocytes, by Chang et al. [10] on DU-145 cells, and Khajah et al. [11] on triple-negative breast cancer cells. At micromolar concentrations, the most likely mechanism of ouabain-induced programmed cell death is due to energy substrate depletion. However, mitochondria-mediated apoptosis by increasing intracellular Ca^2+^ concentration remains a mechanism to be discussed, but requires further investigation [27,28,29,30]. The viability was not significantly affected by ouabain treatment for MSCs nor TAFs. 

The cell cycle is influenced by the action of ouabain in SK-BR-3 cells. We also observed an increase in the level of apoptotic cells (in the sub-G_1_ phase) and a reduction in the number of cells in the synthesis phase (S) following exposure to different ouabain concentrations. These results complement the data published by Chang et al. [10] on prostate cancer cells describing an increase in the number of cells in the G_0_/G_1_ phase and in the apoptotic (sub-G_1_) phase. The number of cells in the S phase, respectively G_2_/M, remained constant [31]. Hiyoshi et al. [32] showed a reduction in the number of cells in the G_0_/G_1_ phase and an increase in those in the S and G_2_/M phase following exposure to ouabain, 50 nM, for 48 h [31]. Contrary to the above, Khajah et al. found that exposure for 24 h to ouabain did not significantly change the percentage of cells in the G_0_/G_1_ phase [11]. The cell cycle was not significantly impaired in tumor microenvironment cells (TAFs) compared to normal cells (MSCs).

The malignant SK-BR3 cell phenotype also changes upon exposure to the Na^+^/K^+^ pump inhibitor ouabain. Flow cytometry and immunocytochemistry data showed a reduction in the signal intensity of the ouabain-treated cells for vimentin, Her2, CD29, and VEGF-R2. The reduction in vimentin expression was also observed by Liu et al. on the lung cancer cell line A549 following exposure for 15 h to ouabain at 25 nM [33]. Interestingly, a dramatic variation in the expression of characteristic markers was seen in MSCs and TAFs (CD29, CD90, and CD73), but were not correlated with the functional behavior of these cells.

The flow chamber test revealed a reduction in tumor cell adhesion on the VCAM-1 substrate coating the flow chamber. Decreased interactions between cells and VCAM-1 occurred in a dose-dependent manner with respect to ouabain concentration and shear stress generated by the peristaltic pump. This observation is also supported by the flow cytometric investigations that revealed a reduction in CD29 (β1 chain of VLA-4, ligand for VCAM-1) as well as by the studies carried out by Pongrakhananon et al. [34] on β integrin levels following ouabain exposure. Translating these observations into a hypothetical situation of a solid tumor treated with ouabain, the tumor cells will have difficulties invading the adjacent tissues, and even if they manage to reach the circulation, they will lack the ability to form distant solid tumors. Although MSCs and TAFs decreased the expression of CD29 on the cell surface under ouabain treatment, the adhesion to the VCAM-1 substrate was not influenced.

From a structural point of view, the Na^+^/K^+^ pump is a transmembrane enzyme consisting of an α subunit with a catalytic role, a glycosylated β subunit with the role of the assembly and anchoring of polypeptide chains at the membrane level, and a tissue-specific FXYD subunit. Each subunit has a number of isoforms, suggesting differences in the activity and role of isoenzymes across tissues and membrane domains [35]. It is interesting to note that the Na^+^/K^+^ pump is also expressed differently in tumor cells. In the case of glioblastoma [20], clear cell renal cell carcinoma or breast cancer [11], an alteration in expression was observed both quantitatively—increased expression, and qualitatively—the variable expression of isoforms. Another aspect observed in clear cell carcinoma is the under-expression of the β1 subunit [36]. This results in a defect in the implantation of the pump at the membrane, and indirectly in a reduction in the transport activity, similar to a drug blockage. One of the effects of the reduced activity of the pump is the reduced expression of E-cadherins, proteins involved in the formation of tight intercellular junctions that favor the process of tumor invasion [37]. Not only are the tumor cells expressing the Na^+^/K^+^ pump, but also the cells from the tumor microenvironment, which form a metabolically active component of the tumor stroma and shapes the immune response in cancer [38,39,40,41]. We investigated and identified that the expression of different subunits of the Na^+^/K^+^ pump was not significantly influenced following exposure to different ouabain concentrations in TAFs and their normal counterparts, MSCs, but these changes were induced in tumor cells. Variable expression of α1 and α2 subunits of the Na^+^/K^+^ pump, accompanied by a marked decrease in β1 subunit expression, results in an intracytoplasmic increase in the number of α subunits and a reduction in the number of membrane-implanted pumps.

The different patterns of α1 expression we observed on the SK-BR-3 tumor cells can be attributed to a dose-dependent effect of the concentrations of ouabain employed in the study: nanomolar concentrations (10^−^^9^ M, 10^−^^8^ M) do not inhibit ion pump activity but may disrupt signaling pathways mediated by the α1 subunit; micromolar concentrations (10^−^^6^ M, 10^−^^5^ M) inhibit ion pump activity, generating a low intracellular K^+^ concentration. This, in turn, will result in the stimulation of the transcription of the gene encoding the α1 subunit, restoring the balance at the cellular level [42,43].

The α2 subunit is not normally expressed in mammary epithelial cells—it is preferentially expressed in neurons. This may reflect the ectodermal origin of mammary epithelial cells as well as the dedifferentiation and aggressiveness of the SK-BR-3 cell line. 

The β1 subunit is a glycoprotein that ensures the implantation of the Na^+^/K^+^ ATPase complex in the cell membrane. Its overexpression was observed in the control SK-BR-3 cell group. Considering the fact that the Na^+^/K^+^ pump present at the membrane level is crucial for the secondary active transport, responsible for the internalization of nutrients, the overexpression of the β1 subunit in SK-BR-3 cells ensures the supply of nutritional principles necessary for tumor proliferation. In addition to its metabolic role, the β1 subunit of Na^+^/K^+^ ATPase is also involved in the formation of intercellular junctions. α1-β1 Na^+^/K^+^ ATPase dimers ensure, through the β1 component, the extracellular interaction with adhesion molecules belonging to other cells, and through the α1 component, the interaction with ankyrin-1, a protein that anchors the Na^+^/K^+^ ATPase complex to the actin cytoskeleton [44].

## 5. Conclusions

Considering the favorable results regarding the pro-apoptotic and inhibitory effect of ouabain on tumor invasion and metastasis, which are also supported by concordant results in the literature, this study may contribute to the enrichment of knowledge related to tumor biology as well as provide new targets for fighting against cancer.

## Figures and Tables

**Figure 1 biomedicines-11-02205-f001:**
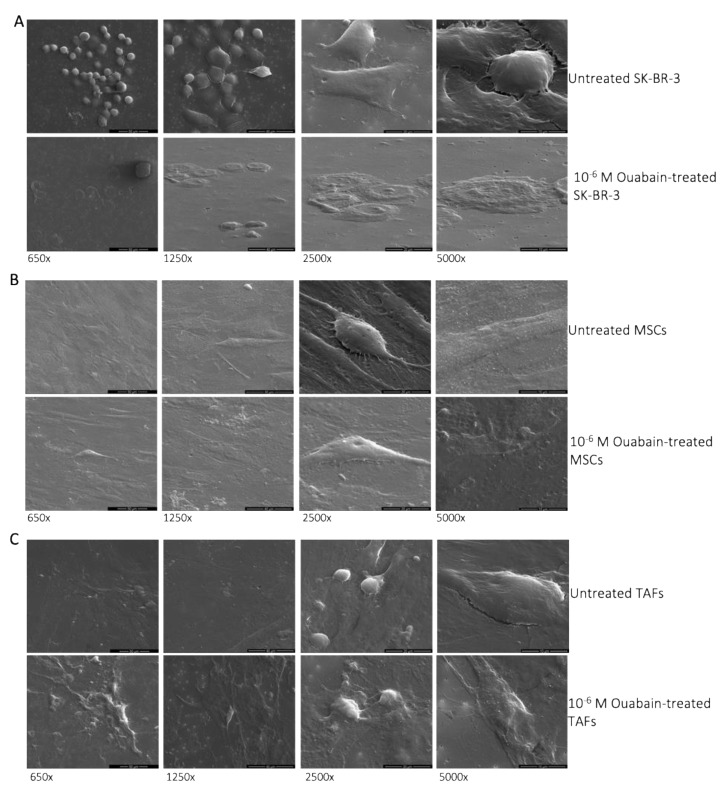
SEM images of the SK-BR-3, MSCs, and TAFs, untreated and treated for 24 h with 10^−6^ M ouabain. (**A**) Untreated control SK-BR-3 cells and ouabain-treated SK-BR-3 cells; (**B**) untreated control MSCs and ouabain-treated MSCs; (**C**) untreated control TAFs and ouabain-treated TAFs. Different magnifications of the SEM images are presented (650×, 1250×, 2500×, and 5000×) to show the morphological changes induced by ouabain treatment compared to the control cells.

**Figure 2 biomedicines-11-02205-f002:**
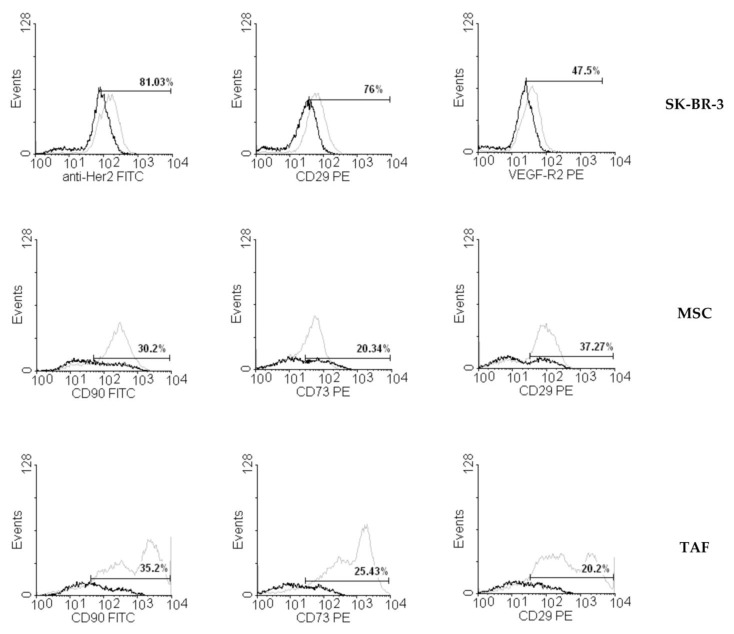
Changes in the expression of surface markers on SK-BR-3 cells, MSCs, and TAFs treated with a 10^−6^ M concentration of ouabain for 24 h. Ouabain-treated SK-BR-3 cells showed only small variations in the expression of characteristic surface markers Her2, CD29, and VEGF-R2, while stromal markers CD90, CD29, and CD73 were significantly decreased in ouabain-treated MSCs and TAFs compared to the control cells. Untreated cells are represented in grey, while the ouabain-treated cells are shown in black.

**Figure 3 biomedicines-11-02205-f003:**
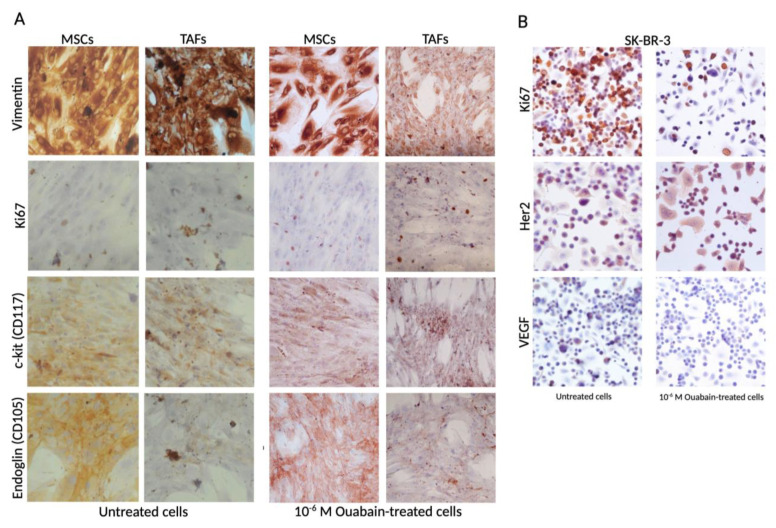
Immunocytochemical evaluation of the MSCs, TAFs, and SK-BR-3 cells. (**A**) Comparative analysis of mesenchymal markers vimentin, c-kit (CD117), and endoglin (CD105) and the proliferation marker Ki67 in the untreated MSCs and TAFs and 24 h after exposure to 10^−6^ M ouabain in the culture media; (**B**) SK-BR-3 cells, untreated and 10^−6^ M ouabain-treated, showing decreased expression in the proliferation rate (Ki67), Her2, and VEGF markers. Ob. 20×.

**Figure 4 biomedicines-11-02205-f004:**
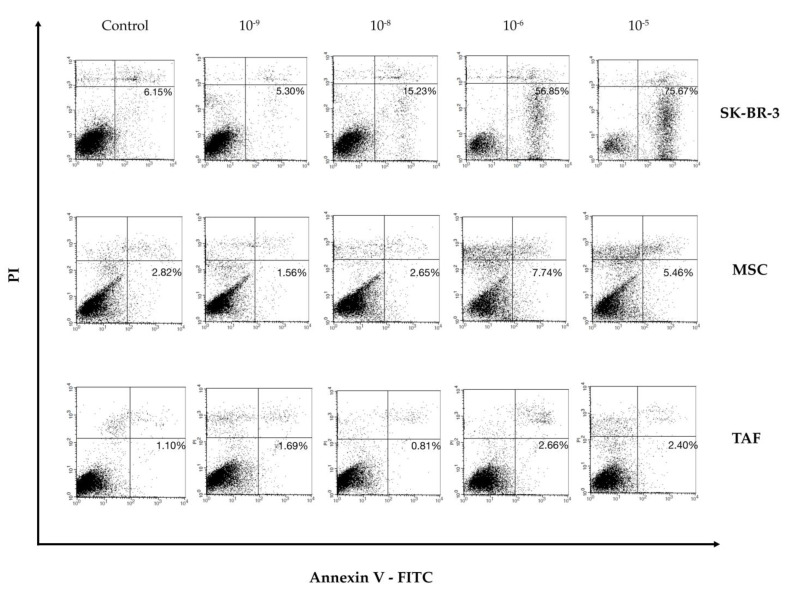
Annexin V/PI assay showing ouabain dose-dependent induced apoptosis in the SK-BR-3 cells, MSCs, and TAFs. The most severe pro-apoptotic effect was observed for the SK-BR-3 cells, with MSCs and TAFs undergoing less stress. A percentage of MSCs and TAFs was positive for PI only, also in a dose-dependent manner, indicative of necrotic death.

**Figure 5 biomedicines-11-02205-f005:**
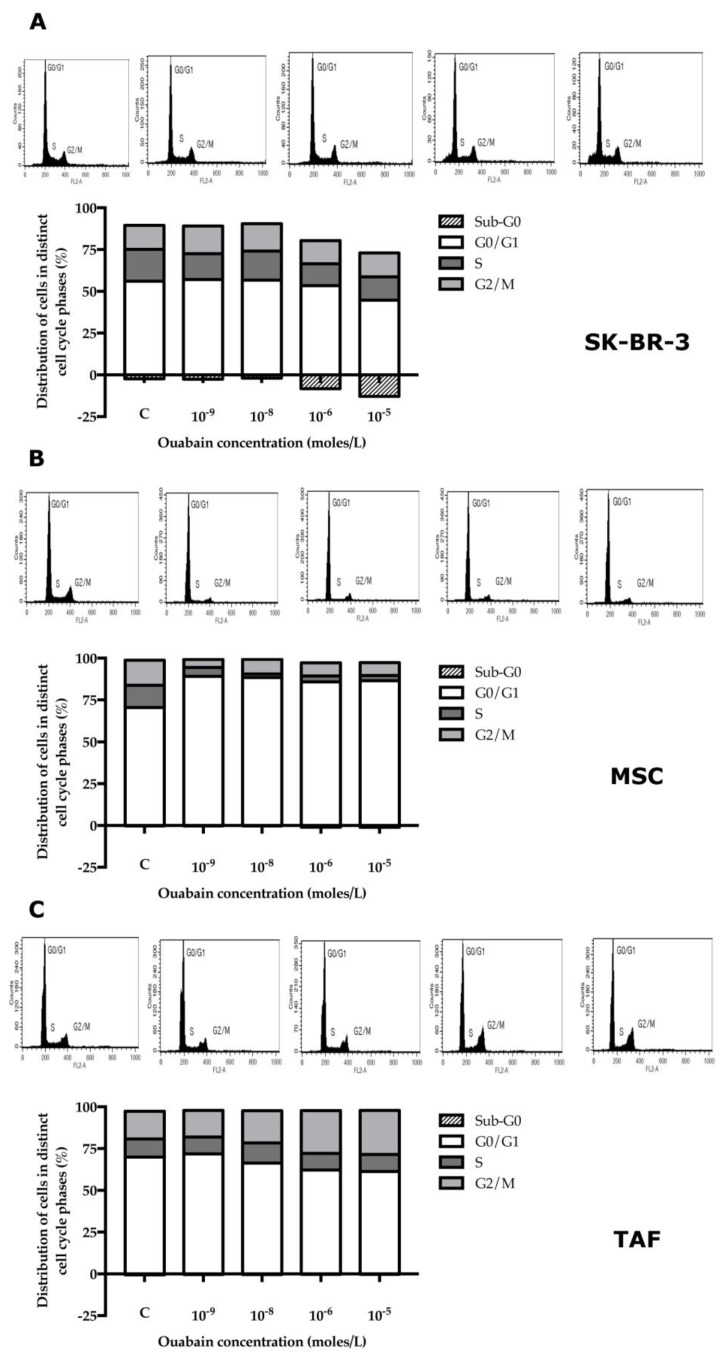
Evaluation of the cell cycle phases in the control and 24-h ouabain-treated cells. All cellular types presented a decreased proportion of cells in G_0_/G_1_ and S phases of the cell cycle, while there was no significant variation for the G_2_/M phase between the control and progressively increasing ouabain concentrations. (**A**) SK-BR-3 cell line; (**B**) MSCs; (**C**) TAFs.

**Figure 6 biomedicines-11-02205-f006:**
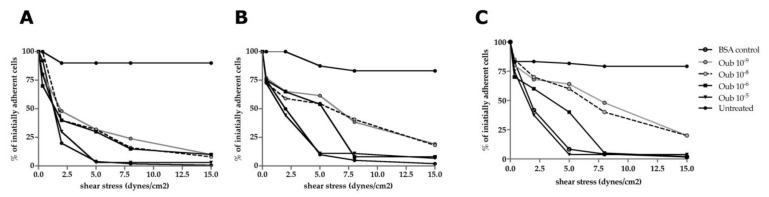
Shear-stress-dependent functional assay of SK-BR-3, MSCs, and TAFs in a flow chamber coated with VCAM-1. A comparative analysis was performed between the untreated cells and cells treated with ouabain for 24 h. (**A**) SK-BR-3 cells; (**B**) MSCs; (**C**) TAFs.

**Figure 7 biomedicines-11-02205-f007:**
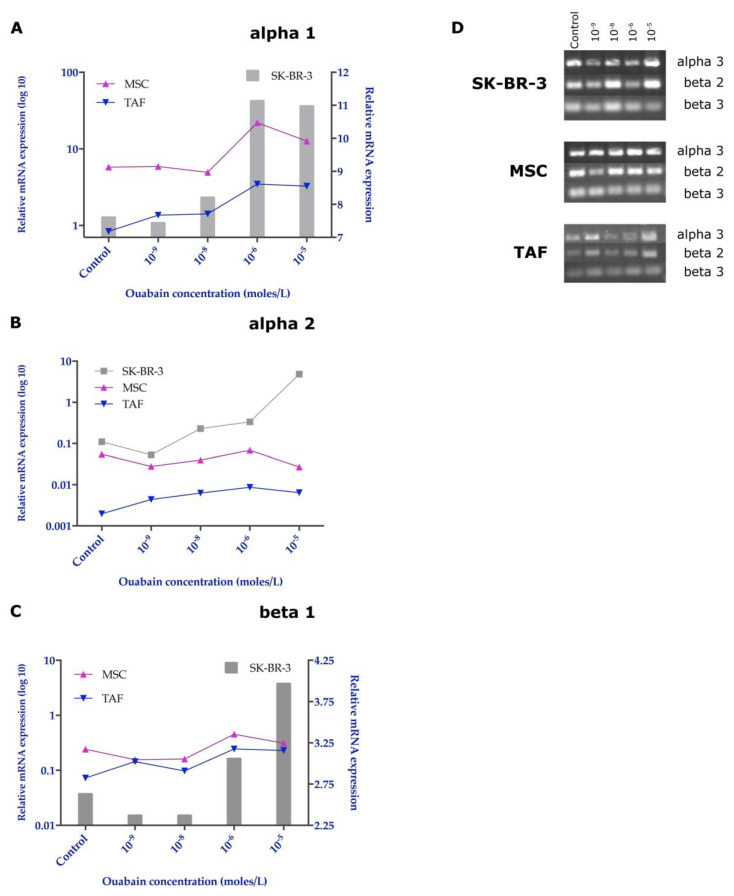
Comparative expression of the Na^+^/K^+^ pump subunits in the control and 24-h ouabain-treated cells (SK-BR-3, MSCs, and TAFs). (**A**) qRT-PCR for α1 subunit expression; (**B**) expression of the α2 subunit; (**C**) expression of the β1 subunit of Na^+^/K^+^ pump; (**D**) agarose gel electrophoresis of RT-PCR amplification products for the α3, β2, and β3 subunits of the Na^+^/K^+^ pump.

**Figure 8 biomedicines-11-02205-f008:**
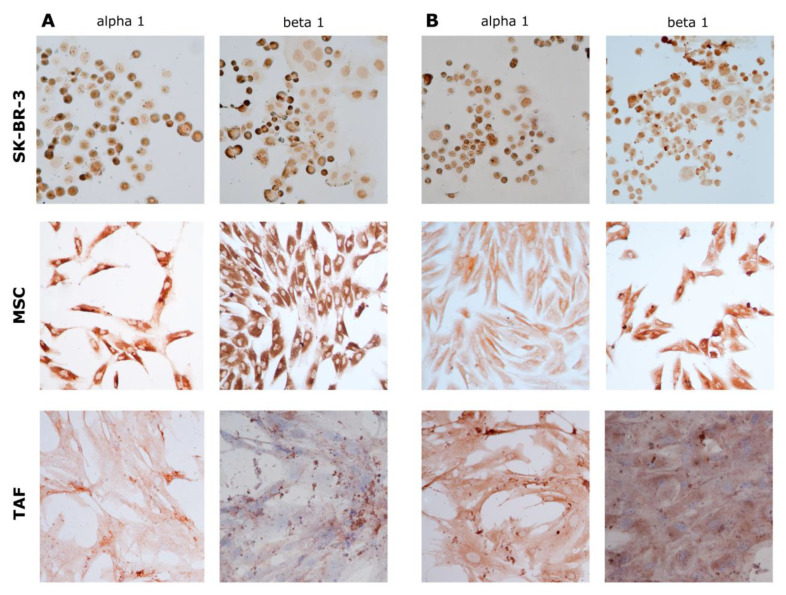
Immunocytochemical evaluation of the Na^+^/K^+^ pump subunits α1 and β1. (**A**) Control, untreated cells showing increased basal expression of the α1 and β1 subunits, especially in the SK-BR-3 cells and MSCs; (**B**) 24 h 10^−^^6^ M ouabain-treated cells. Ob. 20×.

**Table 1 biomedicines-11-02205-t001:** Primers used for RT-PCR.

Na/K-ATPase alpha 1	Forward 5′-AAAAACATGGTCCCTCAGCAA-3′ Reverse 5′-CCACAACTTCCTCCGCATTT-3′	NM_000701.7 (tr. Var 1) 76 bp
Na/K-ATPase alpha 2	Forward 5′-GAATGAGAGGCTCATCAGCATG-3′ Reverse 5′-CAAAGTAGGTGAAGAAGCCACCC-3′	NM_000702.3 77 bp
Na/K-ATPase alpha 3	Forward 5′-AATGCCTACCTTGAGCTCGG-3′ Reverse 5′-CTCGGGCAGGTAATAATGGC-3′	NM_152296.3 69 bp
Na/K-ATPase alpha 4	Forward 5′-GATGATCACAAATTAACCTTGGAAGA-3′ Reverse 5′-TTTGCCCTTTGGTGGCTATG-3′	NM_144699.3 (tr. Var 1) 83 bp
Na/K-ATPase beta 1	Forward 5′-TCAGTGAATTTAAGCCCACATATCA-3′ Reverse 5′-CTTCTGGATCTGAGGAATCTGTGTT-3′	NM_001677.3 74 bp
Na/K-ATPase beta 2	Forward 5′-CCAGCATGTTCAGAAGCTCAAC-3′ Reverse 5′-GCGGCAGACATCATTCTTTTG-3′	NM_001678.3 79 bp
Na/K-ATPase beta 3	Forward 5′-CTGGCCGAGTGGAAGCTC-3′ Reverse 5′-GGTGCGCCCCAGGAA-3′	NM_001679.2 60 bp

## Data Availability

Data are available upon reasonable request to the submitting author.

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
