# Peer review of "In Vitro Study of the Multimodal Effect of Na+/K+ ATPase Blocker Ouabain on the Tumor Microenvironment and Malignant Cells"

_biomedicines, 2023, doi:10.3390/biomedicines11082205_

Round 1

Reviewer 1 Report

The authors describe some effects of the well-known Na+/K+ ATPase inhibitor ouabain on the SH-BR-2 breast cancer cell line, mesenchymal stem cells and tumor-associated fibroblasts. Ouabain is well known for being cytotoxic to the heart and other tissues, and therefore may be a lethal drug. This needs to be addressed. The English needs to be edited, especially in the result section. There are several inaccuracies in the text. Please ensure that all text is concise.

The title is misleading that no tumor microenvironment was studied. This is an in vitro study of three different cell types exposed to ouabain, but no co-cultures. Also, there are only causal data and no mechanistic data. The authors ought to add data showing combined effect of ouabain with other anti-cancer compounds.

The authors need to address what is novel in their study.

Abstract

It lacks the aim of the study and the result section. The method section can be made much shorther.

Line 18: Instead of "cellular cycle" Please write "cell cycle"

Line 24: correct to "flow chamber"

Introduction

Lines 33, 55, 66, 76 and several other places: The + and 2+ should be in superscript.

The first section of Introduction lacks references that should be added.

Lines 38-40: Please specify how the Na+/K+ ATPase intervenes with growth, cell division and apoptosis.

I suggest instead of writing "under-expression" to write: "reduced expression".

Line 56: You need to specify and describe in more the cells in the tumor microenvironment that express the Na+/K+ ATPase pump.

Line 61:  You cannot say "remains increased" – I think you meant "becomes increased".

Line 69: You state that cells (please specify which cells) undergo growth arrest, then of course there is fewer dividing cells. Thus, the last part of the sentence stating "reduced mitotic potential" is superfluous. A reference should be provided in line 71.

Section 72-80: Has ouabain been shown to increase CTL responses in tumor microenvironment? Please add reference to each sentence of the section.

Line 81: Remove "also". You cannot write "In addition" and "also" in the same sentence.

The introduction is too general, and should rather provide specific descriptions of the issues discussed. For instance "involved in the processes of growth, cell differentiation, angiogenesis, migration, adhesion and invasion" in lines 87-88. Please describe which effects are actually seen when Src is activated. Src activation is often associated with increased tumor growth.

Line 90: Which "compensators". Please specify.

Reference is lacking for sentence in line 89-90.

Abbreviations should be written in full first time mentioned (e.g., ATM, NKA).

Line 93: The word NKA comes as a sudden without any early decriptions.  Please provide more information.

Material and Methods

Line 105: The ATCC number should be added.

Line 107: You need to specify the final concentration of penicillin and streptomycin (instead of writing 1% of the stock solution).

The tissue source of the MSCs should be mentioned as well as the catalog number of Lonza.

The tumor source for the TAFs should be stated as well as the exact method description including tissue integration.

Is the FCS heat-inactivated (needs to be stated).

Line 118 and other places: The degree signal should be used in proper way.

Line 120: The Sigma catalog number should be stated (They have many versions).

Line 134: The time of incubation with Triton X-100 and its solution composition (ddw or PBS?) should be stated.

Line 142: The antibody numbers should also be stated here.

Lines 143-145 should provide a better description of the procedure. The concentration of each antibody and the solution used should be stated. The source of the secondary biotinylated antibody should be stated. The source of hematoxylin should be stated.

Line 148 and previous lines: What is the concentration of EDTA?

Line 157: I think you meant "subjected" and not "submitted".

Line 158, 164, 189 and many other places in the text: Add a space between "Flow" and "cytometric".

Lines 159: Remove 24-hours.

Line 160: Add to the end of the sentence: "for 24 hrs". (The same for other places in the manuscript).

Line 168: Correct to: "The cells were washed with".

Line 182: correct to "flow cytometer".

Lins 184-185: add nm to the wavelengths.

Section 2.5: For cell cycle analysis you need to digest RNA, since PI binds to both RNA and DNA – so please correct the text accordingly.

Lines 191,192 and other places in the text: correct to "Flow chamber"

Line 194: Please state in which solution was VCAM-1 dissolved.

Line 197: correct to "adhere to"

Result section

Extensive editorial and scientific editing is required for this section. Each section should start with the question asked, why was the assay done, and then presenting the data.

Line 243: specify the cells.

Line 245: You cannot say "ghost cells such as SK-BR-3". In the current format, the text says that SK-BR-3 are ghost cells. Therefore, rewrite the sentence. Figure 1: Please provide also lower magnifications to show more cells. The images presented are with different magnifications. Please use uniform magnification of the images. Please also provide light microscope images. The surface of TAFs (and the number of cells) are also altered. So the text should be rewritten.

Line 245: correct to SK-BR-3.

Figure legend 1 should state the times magnification. The bars should be made more visible.

Line 254: correct to "Flow cytometric"

Section 3.2: English editing required. THe mean fluorescent intensity should also be stated. The percentage is a matter of what you determine as positive. ± values should be provided. The staining with 2nd antibody should also be shown. The % expression of control cells should also be shown in the figure. What happens to SSC/FSC ? The numbers of lines 260-261 should rather be added to a summary Table of Figure 2.

Lines 268-269: English editing is required.

Are there markers that are upregulated?

Section 270-276 should refer to the Figure.

Please also describe why the different markers were studied.

Line 274: Does 25% refer to MSCs or TAFs? Also, the standard deviation should be added.

Line 270: Decide whether to write hours, hrs or h throughout the manuscript.

Section 3.2 – There are many jumps in the text. Please reedit to make it more structured and organized. Describe each cell line separately, and also emphasize what you want to tell with your data.

Figure 3: Should show the magnification. The ouabain images for MSCs and TAFs are lacking. Why are there so many round cells in SK-BR-3? Quantification of the staining intensities should be done.

Line 293: correct to: "flow cytometric" (Instead of flow cytometric analysis you can write "flow cytometry").

Section 3.3 should be rewritten. In the current text it says that "flow cytometric analysis causes an increase" while it is ouabain that increases the expression.  There are also several editorial issues in this section (extra spaces, correct to superscript etc).

The labels of the arrows in Figure 4 are wrong and also their directions (Annexin V-FITC is from left to right, and PI from the lower part to the higher part. The "cell counts" doesn't belong – it is the fluorescence intensities.

In the legend, please mention how many events were collected. And make an accompanying table for the data.  Also, the percentage of PI positive cells should be stated and double stained cells.

The 1e-09, 1e-08 etc should be corrected to scientific values in the figures.

Line 306: "most" is not a correct word here. You have even more with 10-5M. Maybe write "significantly".

Refer to the Figures when describing in the text. There are sections lacking the reference to Figures.

Please correct to G0/G1, G2/M, sub-G0

And put attention to all what should be in sub or superscript.

Line 311: The concept "pre-apoptotic SK-BR-3 cells (in the G0 phase)" is wrong. Please rephrase. Apoptotic cells appear in Sub-G0 (this is not a cell cycle phase).  Please let a scientist check the right use of scientific words.

Line 315: "sub-G0 phase, when cells exit the cell cycle." This is not right. Sub-G0 cells are cells with less than 2N DNA due to fragmentation of the DNA, not because of an exit of cell cycle.

Please show cell death also by staining with anti-cleaved Caspase 3. This staining can be combined with cell cycle analysis.

Line 327: a space after "flow" and all other places.

Line 330: You say: increasing shear stress from 0.35 dyne/cm2 – but until when and what are the steps? Was a lower shear stress force also applied? The text should be corrected by stating the range.

Section 3.4 lacks time, and reference to Figure and there are several inaccuracies. It says >15%, then reduction from 98.37% to 41.42% which is much more than 15%. Please be accurate.

Lines 338-342 – Which tumor adhesion molecule can interact with VCAM-1? These should be studied.

Figure legend 6 should describe how the cell number was determined.

All figures lack statistics. Also, the method section lacks a section of statistical analysis.

Figure 7 – It is unclear why there is a mixture of bars and lines. Better if all data are in bars.  The supplementary figure is exactly as Figure 7D. The authors should provide the entire image of its subfigure. The time of incubation should be stated. What happens at earlier time points before the cells die?

The manuscript requires extensive English as well as scientific editing.

Reviewer 2 Report

In this manuscript, the authors compared the responses of three cell lines (breast cancer, stem cell, and fibroblast) to Na+/K+ ATPase blocker Ouabain treatment and concluded that Quabain affects the tumor microenvironment. The conclusion is vague, although the study topic is interesting. Other concerns include:

Abstract: it needs to be rewritten to summarize the results, not the methods.

Introduction: This is too long. The last three paragraphs can be deleted.

Figure 2: the grey line is unclear, suggest using two different colors.

The sequence of data presentation can be reorganized, such as by moving Figures 2 & 3 after Figure 5, as the potential mechanisms of action.

A diagrammatic figure is suggested to be added to compare the different reactions to the drug among the three cell lines. The Discussion needs to be rewritten based on the diagrammatic figure.

English is fine.

Reviewer 3 Report

* The authors investigated the effects of Oubain on tumor development. I find the topic interesting but some issues must be addressed.

* Line 25-27: the conclusion must be more clear and address the effect of Oubain on the different cell types.

*  The introduction is so long. Please, shorten it and transfer non-necessary information to the discussion section. 

* Please include all catalog numbers for all used kits in the materials and methods section.

* Line 104: could the authors explain why did they use a breast cancer cell line? why did not use cell lines for other types of cancer?

* Line 136-140, please list all used antibodies in a table including their sources, concentrations, and catalog numbers.

* Line 145: please, mention the used camera and microscope. I suggest making a quantitative analysis using image J and statistical analysis for many fields to reveal the differences between investigated cell lines. 

* Line 212-223: Please list all used primers in a table.

* Line 241: Please, arrange the results section according to the arrangement in the materials and methods section.

* Line 242: what is the significance of adherence? Did all cells have the same lesions? is it bad or good effects? please, analyze your results using the chart.

* Line 277: The average of the Ki67 values was 61.68% for the untreated SK-BR-3 cells. How did you measure?

* Figure 3A: where are the untreated groups for MSC and TAF? and where are the Endoglin results for the cancer cell line?

* Figure 7D: Please, express the results by a chart.

* Figure 7: I suggest using western blot for quantitative measuring of proteins 

* Figure 8: Please, express the results by charts.

* The writing style requires more modification by a specialized English editing company. There are some Grammar mistakes and some repeated abbreviations in the main text such as TAFs in lines 100 and 111.

Author Response

Dear reviewer,

Thank you very much for the reviewing of the paper. Your opinion was very valuable to us, and we were grateful for all the suggestions you made.

We corrected the typing errors and scientific interpretation mistakes.

We included the catalog numbers for the reagents and kits used in the experimental part.

We added Supplementary materials – images of cells in optic microscopy and electron microscopy.

We quantified the PCR gels in ImageJ and provided analysis graphs in Supplementary materials.

We re-wrote the Abstract, Introduction, Results and Discussion parts.

We had English proof reading for the entire paper.

Thank you very much!

Reviewer 4 Report

The aim of this manuscript is to investigate the possible anti-cancer effect of ouabain at a lower dose in several cancer cell lines. This study is interesting in this field; however, there are several major issues that need to be addressed before it can be published.

1. Introduction Line 81-88: Please include Dr. Zijian Xie's initial work as he was the founder of this concept. For example: (Xie et al., 2003; Ye et al., 2013)."

2. A toxicity study of ouabain should be conducted on the cell lines to justify the use of 1uM ouabain in the study.

3. Figure 1: Please add labels to indicate which group represents the cells after Ouabain treatment.

4. Figure 3A: The untreated group should be included for comparison.

5. Figure 7: The expression of Na/K-ATPase alpha 3 should also be analyzed and presented.

6. Figure 7: In addition to gene expression, the protein expression of Na/K-ATPase alpha and beta subunits should be investigated and presented.

7. Figure 8: The expression of alpha and beta subunits should be quantified and presented in the figure.

English should be revised. 

Round 2

Reviewer 1 Report

The article has been improved, but still English editing is required.

In abstract, the authors emphasize the potential medical use of ouabain in treatment of cancer, but they should address the cardiotoxic and potential lethal effect of the compound. According to Figure 1, ouabain changes the morphology also of MSCs and the TAFs.

Line 41: The word "sector" is not appropriate and should be exchanged with another word.

Line 72: Kappa should be written with the letter κ.

Line 153: Spelling mistake – should be IU. The 100 IU/ml refers to Penicillin. You also need to state the final concentration of streptomycin.

Line 159 – check if the address of Sigma is right. (They are situated in St. Louis).

Line 205: BD should be written in full name and location.

Section 2.9. The test assay and the replicates should be described.

The supplementary data should be better organized with a legend for each figure. In the current state some of the figures are not clear.

Concerning figure 1: It is important that all images are in the same magnification, and both lower magnification (providing a panoramic view on the cells) and the higher magnification to see the individual cells should be shown. 

Line 347 – Spelling mistake – correct to ouabain.  (There is a P instead of o).

How can you explain the strongly reduced expression of the surface markers by ouabain?

Figure 3A lacks the untreated cells MSCs and TAFs. (According to figure legend – these images represent ouabain treated cells only).

Figure 4 – The percentage of PI+ cells should also be stated (which are increased in ouabain-treated MSCs). An MTT cytotoxicity test should also be performed for the three cell types.

How can you explain the increased expression of the Na+/K+ pump subunits?

The English requires editing by a scientific English editor.

Author Response

Dear reviewer,

Thank you very much for reviewing our paper. Your opinion was very valuable to us, and we were grateful for all the suggestions you made.

Reviewer 2 Report

Authors have addressed my concerns, no further comments.

Author Response

(The authors gave the same response as above.)

Reviewer 3 Report

* I have no more comments 

Author Response

(The authors gave the same response as above.)

Reviewer 4 Report

Some of the Previous reviewer’s comments are not properly addressed.

Previous comments

#1. More reference from Dr. Xie should be added

#4. The normal control should be added to the main figure not supplement, which is very important for the readers

#6. No protein expression was added on Figure 8, Western blot or Elisa should be done with quantification.

New comments: Please organized your supplement data and cited in manuscript as Figure S1, Table S1….

Author Response

(The authors gave the same response as above.)

Round 3

Reviewer 1 Report

Comments to ouabain version 3

The English editing should be done by a scientific editor that understands the topic. Several of the sentences have become impossible and inaccurate.

Line 20-21: I would suggest correcting to: "determining the cytotoxic effects of Ouabain."

Line 48-49: Please mention which proteins the pump interacts with. Instead of writing "intervening with" describe exactly which processes are influenced by the pump. In this paragraph, also mention the already known effects of Ouabain in cancer.

Paragraph 66-74: If inhibition of the Na+/K+ pump leads to the activation of pro-survival pathways and cell proliferation (as it says in the text), how can it be that Ouabain that inhibits this pump does the opposite? Could it be that you had another intention in the paragraph? Please verify that the text is right.

Line 75: "This" refers to what has been described in previous sentence, but in the previous sentence the antiproliferative and cytotoxic potential was not described. Please rewrite the section to be accurate.

Also, the next sentence is complicated: If it induces death of immune cells, how can it improve the anti-tumor activity of immune cells?

The introduction has to provide accurate information.

(In next version, please show the corrections done to version 3, without the corrections of version 1 and 2).

Line 315: you can not say "resistance" to a concentration. The cells do respond to these concentrations, so they are not resistant. The sentence has to be rewritten.

You cannot say "in the range of" if it points to one concentration. Just say, at 10-6, etc.

Figure 1: The higher magnifications are not from the same field as of the lower magnification. This is quite confusing. For instance, A control – The reader expects a gradual increase in magnification, but there is no relationship between magnification 2500x and 5000x. For the ouabain-treated samples the 650x magnification is not related to the others. Please be sure about the images. Also correct for B and C images.

Line 353: Spelling mistake – correct P to O.

Figure 2 lacks the unstained samples.

In Title of 3.2: Instead of "healthy cells" write MSCs.

The concept "immunophenotypic changes" is misleading. You didn't study the immune cells in vivo, but changes in surface markers on the stated cells.

There is also a need to describe why these markers were studied.

Many of the sentences need to be rewritten for better presentation. For instance line 411: "Ouabain treatment….. treated with Ouabain". Please go over the manuscript to be sure that the sentences are right before you resubmit.

Line 423: You can't say "arrested in a sub-G0 phase" They can appear as sub-G0 cells. (They can be arrested in G1 and G2).

Most of the results deal more about the concentrations of ouabain rather than the effect of ouabain. The authors should focus on the latter instead of the former.

Line 473: spelling mistake. Correct to "prominent".

Use another word than "plasmalemmal".

Why is the PCR done on other subunits than the immunostaining?

The Supplementary data have been improved, but still the legend need to provide a more detailed description. Supplementary Figure S1: Should state the incubation time with ouabain, and the antibody used. The size marker should be shown.

Supplementary Figure S2: The different labels should be described in the legend: Define: 3000C-C1-4, 3000C, etc.

Supplementary Figure S3:  The concentration should be written with superscript where needed. And Ouabain and M should be included in X-axis. Time of incubation should be stated.

English and scientific styling are required.

Author Response

Dear reviewer,

Thank you very much for the reviewing of the paper. Your opinion was very valuable to us, and we were grateful for all the suggestions you made.

Reviewer 4 Report

Protein expression, such as western blot or Elisa are invaluable as compared to qPCR in protein expression. Na/K-ATPase can be detected by Western blot or Elisa. 

Plus, Pumping function of Na/K-ATPase should be tested if possible, which is also very important in this field. 

Author Response

(The authors gave the same response as above.)

Round 4

Reviewer 1 Report

Comments to revised ouabain version 14th July 2023

The authors have not addressed all the major concerns of my previous comments.

Concern 1: The title provides an impression that this is an in vivo study ("tumor microenvironment" is an in vivo concept). But the authors only studied three cell types in vitro (each cell type separately, and there is no experiment even mimicking the tumor microenvironment such as in vitro co-cultures), such that the title should emphasize that this is an in vitro study.

Concern 2: Introduction (lines 54-62). I asked the authors to better describe the issue that "blockage of the Na+/K+ pump leads to cell proliferation", which actually contrasts the described anti-proliferative and cytotoxic action of its inhibition. The authors need to address this contradictory issue in more depth. Is this dose-dependent? Is this a mechanism to compensate for the cytotoxic effects? In which cell types are the proliferative signal pathways activated by ouabain and at which concentrations?

Concern 3: The Menger paper [15] should be described in more detail in order to introduce the readers to the whole picture and not to a fragment citation (and use your own words, and not only copy the words of the paper).

Line 177: A space between flow and chamber should be added.

Concern 4: Line 229: When I provided a comment, it is because the text has a scientific conceptual problem, and the authors ought to try to reformulate the sentence to become more concise. The word "resistance" is problematic in this context (especially if you show in Figure 4 that more than half of the cell population undergoes apoptosis). (Every sentence needs to be precise). I would instead write: "At 10-5 M or 10-6 M, ouabain….".

Line 234: Instead of writing "subjected to the action of concentration of 10-6 M Ouabain for 24 hours" I would suggest writing " subjected to 10-6 M Ouabain for 24 hours".

Concern 5: Result section: The SEM pictures only provide information of a very small field. Light microscopy images should be provided in parallel.

Concern 6: The authors state that "unstained samples will not provide more information". Unstained samples are important to determine whether the low intensity population is due to a downregulation of the markers or a loss of their expression. Therefore, it is important to show the unstained samples.

Line 266: The wording "was retained" is not correct in the context used, and must be corrected to a proper word. E.g., "expressed" (Ki67 is staining proliferating cells). The word "only" is superfluous.

Lines 272-273: Instead of " could potentially have an important antiproliferative effect" you should write " has an antiproliferative effect".

Figure 3: Why is the staining in some pictures yellow and in the other brown (the expected staining color)?

Line 289 and other places: What is the significance of 2 digits after the comma? Each time a % number is provided the ± std should be included.

Lines 298-: The viability of MSCs was only slightly reduced. So the sentence should be rephrased.

Please be consistent in the text and figures to write G0/G1 with the numbers in subscript.

Line 313-314: Instead of " of the blocker targeting the Na+ /K+ pump" please write "Ouabain".

Line 331: Instead of: "exposed to a range of decreasing concentrations of " (it might likewise be "exposed to increasing concentrations…..), you can simply write: "exposed to various concentration of…".

Lines 333 and 348 and other places…….: Please add a space between "flow" and "chamber".

Figure 6. It would be easier for the reader if the different lines were in color. The control here should be uncoated flow chambers. This is especially important in the light of the first sentence of the result section – that ouabain causes the tumor cells to become more adherent…..

Line 353: There is no need to say "in the culture medium" (the cells are exposed to ouabain!).

Ensure that the time of incubation should be stated in all figure legends.

As I previously asked for: All three cell types should be presented in bars in Figure 7A and C.

Lines 423 and 427: Instead of "pre-apoptotic" it should be written "apoptotic cells".

Line 435: A space should be added between "flow" and "cytometry" also other places flow is attached to the next word. Please make a space after every "flow" (e.g., line 444.)

Line 435" The data cannot "describe" they can "show" so please exchange the words.

What can be the reason for the reduced surface expression of indicated genes?

Line 440: Explain in the text what you mean with: " but are not correlated with the functional behavior of these cells" which functional behavior are you referring to?

MSCs have supportive roles for many tissues. How does ouabain treatment affect this function?

Line 466: correct to: "expressing" (remove hyphen)

If Her2 expression is reduced, would ouabain treatment affect anti-Her therapy?

The English has been improved. But many non-scientific phrases have been used.

Author Response

(The authors gave the same response as above.)

Round 5

Reviewer 1 Report

Comments to Ouabain v5

Despite that the authors have not corrected the manuscript according to some essential comments, I will now only give minor comments that have to be done in order to present the data in a sufficient high quality.

Figure 2: The reason I asked for showing the unstained cells is to let the readers know whether the "low fluorescence intensity" population is "low expressing" or "negative". Therefore the unstained should be shown.

The superscripts in the X-axis of Figure 2 should be superscripts and not at the almost same level of 10.

Figure 4: The percentage is only given for Annexin V-positive cells in the quadrants. That of PI+ should also be provided in the quadrants (the upper left quadrant).

Figure 5: The labels of the axes of flow cytometry should be made clearer.

Figure 6: It is almost impossible to distinguish between some of the lines – especially BSA control and Untreated. Both are black solid line with a closed circle. Therefore, the authors must present the lines in a way that makes it easier to distinguish between the groups.

In Figure 6B, the upper line is thinner than the continuation of the line. Please correct.

Figure 7. It is not clear which y-axis is for which cell type. Why cannot all cells be presented as done for 7B?

The English is more and less OK.